



# Pre-training for Deep Statistical Climate Downscaling: A case study within the Spanish National Adaptation Plan (PNACC)

Jose González-Abad[1], Maialen Iturbide[1], Alfonso Hernanz[2], and José Manuel Gutiérrez[1]

[1]Instituto de Física de Cantabria (IFCA), CSIC-Universidad de Cantabria, Santander, Spain
[2]Spanish Meteorological Agency (AEMET), Madrid, Spain

**Correspondence:** Jose González-Abad (gonzabad@ifca.unican.es)

**Abstract.**

Deep Learning (DL) has recently emerged as a promising approach for statistical climate downscaling. In this study, we investigate the use of pre-training in this context, building on the DeepESD model developed for the Spanish National Adaptation Plan (PNACC), which uses ERA5 predictors and the 5km ROCIO-IBEB national gridded predictand dataset. We evaluate the effectiveness of different fine-tuning strategies to adapt this pre-trained model to alternative regional predictand datasets, specifically a point-based station dataset. The objective is to develop downstream downscaling methods that maintain consistency with the original national-scale model while capturing the specific characteristics of regional and local datasets.

We analyze the benefits of fine-tuning in terms of faster convergence, improved generalization, and greater consistency. Using eXplainable Artificial Intelligence (XAI) techniques, we examine the relationships learned by the models and compare the resulting climate change signals. Our results demonstrate that pre-training provides a robust foundation for statistical downscaling, particularly in cases with limited spatial and/or temporal data availability (e.g., local high-resolution datasets available only for short periods), thereby reducing epistemic uncertainty and improving the reliability of future climate projections. Overall, this approach represents a step toward standardizing DL-based downscaling models to ensure more coherent and consistent climate projections across national and regional scales.

## 1 Introduction

Global Climate Models (GCMs) simulate the spatio-temporal evolution of the climate by numerically solving the physical set of equations that govern its dynamics (Chen et al., 2021). GCMs are used to generate future projections along different forcing or greenhouse gas emission scenarios, providing possible future socio-economic pathways (Eyring et al., 2016). However, due to inherent physical and computational limitations, the resulting projections have a coarse spatial resolution, which limits their suitability for regional studies. Statistical downscaling (Maraun and Widmann, 2018) addresses this limitation by employing statistical models to learn the relationship between coarse large-scale variables (predictors) and the local variable (predictand) of interest (Gutiérrez et al., 2019).

Recently, Deep Learning (DL) (Goodfellow et al., 2016; Prince, 2023) has emerged as a powerful tool for statistical downscaling, thanks to its ability to model non-linear relationships and effectively process spatial data. As a result, DL techniques



have been applied to a wide range of statistical downscaling problems, from simple super-resolution (Vandal et al., 2017; Sha et al., 2020a, b) and bias adjustment (François et al., 2021) methods, to more sophisticated Perfect Prognosis (PP) (Baño-Medina et al., 2020, 2022) and emulation (Doury et al., 2023, 2024; Baño-Medina et al., 2023) approaches which rely on large-scale synoptic predictors—reliably simulated by GCMs—to learn empirical relationships with regional or local variables of interest.

Deep PP downscaling methods (hereafter deep downscaling) have already been used to produce regional climate change projections in various regions (Baño-Medina et al., 2021, 2022; Soares et al., 2023; Balmaceda-Huarte et al., 2024). For example, the new generation of regional climate change scenarios for the Spanish National Adaptation Plan, based on CMIP6 (Escenarios-PNACC 2024), includes a deep learning downscaling method (DeepESD) developed using ERA5 predictors and a 5 km gridded observational dataset over Spain (González-Abad and Gutiérrez, 2025). These downscaled scenarios serve as

the primary source of information for developing impact assessments and adaptation studies in Spain. However, other regional scenario datasets have also been produced for specific applications or sub-regions, using alternative high-resolution grids or point-based station data with different downscaling techniques (Monjo et al., 2016; Amblar-Francés et al., 2020; Miró et al., 2021; Hernanz et al., 2022). Such methodological diversity can produce divergent outcomes, which may confuse end users. The possibility of using a baseline downscaling model that can be adapted to new regional datasets (high-resolution grids or point

observations) would facilitate the generation of downstream scenario products and improve their consistency. In addition, having a model with pre-learned relationships could be especially valuable in low-data regimes, which are common when working on specific regional scenarios where the amount of available data is often limited.

One promising approach to achieve this is to rely on pre-training (Bengio et al., 2006; Vincent et al., 2010; Erhan et al., 2010). In this paradigm, a DL model is first pre-trained on an initial dataset(s) and then fine-tuned on one or several target

datasets. This allows the model to learn meaningful, general-purpose representations during the pre-training phase, which can then be effectively transferred to tasks related to those of the original dataset. For instance, much of the recent success of large language models can be attributed to an extensive pre-training phase involving massive amounts of data (Radford, 2018; Kenton and Toutanova, 2019; Brown, 2020). These pre-trained models are then fine-tuned for specific tasks, such as following human instructions accurately. Similarly, in computer vision, pre-training has been instrumental, enabling fine-tuning of models that

provide rich representations of spatial features (Dosovitskiy, 2020; Radford et al., 2021; Caron et al., 2021; He et al., 2022). More recently, in weather and climate science, large deep models have been trained with large combinations of datasets to provide foundational models that capture a wide range of physics-based phenomena inherent in these systems (Nguyen et al., 2023; Lessig et al., 2023; Bodnar et al., 2024; Schmude et al., 2024).

In this study, we investigate, for the first time, the use of pre-training in the context of deep statistical downscaling.

Specifically, we build on the DeepESD convolutional model originally developed for the Spanish National Adaptation Plan (PNACC), which downscales coarse-resolution GCM predictors to a high-resolution (5 km) national gridded observational dataset (González-Abad and Gutiérrez, 2025). Using this pre-trained model as a baseline, we explore the effectiveness of various fine-tuning strategies to adapt it for alternative national or higher-resolution regional predictand datasets. The aim is



to develop cost-effective downscaling methods that maintain consistency with the national-scale results while capturing the specific characteristics of local datasets.

We focus on daily minimum and maximum temperatures and precipitation. To test the fine-tuning performance, we employ an alternative point-based observational dataset comprising over 3,400 stations for temperature and 5,800 stations for precipitation across peninsular Spain. This station-based dataset serves as a compelling test case due to its distinct resolution and nature compared to the original gridded data. To assess the consistency and interpretability of the fine-tuned models, we analyze the learned relationships using eXplainable Artificial Intelligence (XAI) techniques (Adadi and Berrada, 2018; Arrieta et al., 2020; Minh et al., 2022; González-Abad et al., 2023). Additionally, we evaluate the benefits of pre-training in terms of faster convergence and improved generalization during the fine-tuning process.

The paper is structured as follows. Section 2 introduces the data, the DL model, and the XAI techniques used in this work. In Section 3, we present the pre-training and fine-tuning strategies in detail. Section 4 provides the results of all the experiments conducted in this study. Finally, in Sections 5 and 6, we discuss these results and conclude with the main findings of this work.

## 2 Experimental framework

In this section, we first describe the region of study, the datasets, and the preprocessing procedures. We then provide a detailed overview of DeepESD, the deep downscaling architecture that serves as the foundation for this work. Finally, we introduce the techniques used to assess the deep downscaling models, focusing on the relationships they learn.

### 2.1 Region of Study

We focus on peninsular Spain (36°N–44°N, 9.5°W–3.5°E), which represents a challenging benchmark for statistical downscaling due to its diverse climatology and complex orography. This region, located within the Mediterranean basin, is significantly affected by climate change, experiencing increasing temperatures and changes in precipitation patterns (Hoerling et al., 2012; Cos et al., 2022).

In this region, multiple observational datasets are available, including several gridded datasets such as ROCIO-IBEB ($5\,km$ resolution Peral García et al., 2017), Iberia01 ($10\,km$ resolution Herrera et al., 2019) and E-OBS ($10\,km$ resolution Cornes et al., 2018), as well as global higher-resolution grids at $1\,km$ (Karger et al., 2023) and over specific sub-domains (Basque Government, 2020; Taboada et al., 2024). This provides opportunities to extend this work using multiple datasets of different natures for pre-training and/or fine-tuning.

### 2.1.1 Predictor and Predictands

As predictors, we select a set of large-scale atmospheric variables commonly used in previous climate downscaling studies (Gutiérrez et al., 2013; Baño-Medina et al., 2021; Soares et al., 2023), specifically air temperature, specific humidity, and meridional and zonal wind velocity at 850, 700 and 500hPa and mean sea level pressure. These predictors are obtained from the



ERA5 reanalysis dataset (Hersbach et al., 2020) and regridded from their original 0.25° resolution to 1.5° using conservative
interpolation, to match the coarser scales typical of GCM outputs.

To ensure that large-scale phenomena influencing the downscaled variables are fully captured, we extend the spatial domain
to 23.5°N–68.5°N and 39°W–22.5°E. Finally, to avoid biases from differing variable scales, each predictor grid point is
standardized to a zero mean and unit variance before being fed into the model.

For this study, we used two types of observational data from the Spanish Meteorological Agency (AEMET) as predictands:
the ROCIO-IBEB gridded dataset (Peral García et al., 2017), which provides daily precipitation and temperature at 5 km
resolution, and station observations from the STATIONS-IBEB network (Spanish Meteorological Agency, 2021), comprising
over 5,800 precipitation and 3,400 temperature ground stations. ROCIO-IBEB served as the predictand for developing the
deep downscaling model used to generate high-resolution gridded projections under the Spanish National Adaptation Plan
(PNACC). In this study, we pre-train the downscaling model on ROCIO-IBEB and then fine-tune it on STATIONS-IBEB
observations. We also train a separate model from scratch using STATIONS-IBEB as the predictand to benchmark the benefits
of the pre-training approach. We refer to this model as *fully-trained* to indicate that it is trained entirely on STATIONS-IBEB,
without leveraging any pre-trained model on ROCIO-IBEB.

Figure 1 presents both the climatologies of mean and extreme-related indices for each of the three predictands (minimum
temperature, maximum temperature and precipitation) for both datasets. Specifically, we show the mean values of the three
variables as well as the annual minimum of daily minimum temperatures (TNn), the annual maximum of daily maximum
temperatures (TXx), and the annual maximum daily precipitation (RX1day). Regional differences in extremes—driven by
orography and coastal versus inland conditions—are apparent in both datasets. Although the ROCIO-IBEB gridded product
is derived from STATIONS-IBEB observations, the station-based dataset shows higher extreme values, especially for pre-
cipitation. It is also worth noting that the number of available stations for precipitation is higher, as reflected in the denser
precipitation maps for the STATIONS-IBEB dataset.

### 2.1.2 Historical and Future Projections

To evaluate the performance of the DL model to downscale future projections from GCMs, we follow previous studies
(González-Abad and Gutiérrez, 2025) and use the EC-Earth3-Veg climate model (Döscher et al., 2021), which is among
the GCMs recommended by EURO-CORDEX for downscaling CMIP6 over the European domain (Sobolowski et al., 2023),
also used in Escenarios-PNACC (Correa et al., 2023). We use predictor data from the historical (1980-2014) and a future
scenario (SSP3-7.0, 2071-2100) representing high emission forcing conditions. Following the assumptions of the PP approach
and previous studies (Baño-Medina et al., 2021, 2022; Addison et al., 2024), we apply a simple mean-variance bias adjustment
to the GCM predictors to ensure that their distribution more closely matches that of their ERA5 reanalysis counterparts. For
further details on this transformation, we refer the reader to Baño-Medina et al. (2021). Prior to feeding the bias-adjusted GCM
outputs into the deep-learning model, we standardize them using the ERA5 grid-box means and variances.





**Figure 1.** Climatologies for the period 1980–2010 of minimum and maximum temperatures and accumulated precipitation. For each of the three variables, we show both the mean climatology and an extreme-related statistic: the annual minimum of daily minimum temperatures (TNn), the annual maximum of daily maximum temperatures (TXx), and the annual maximum daily precipitation (RX1day). Each variable's climatology is computed for the ROCIO-IBEB and STATIONS-IBEB datasets (arranged in rows within each subpanel).

## 2.2 Deep Learning Model

We select the DeepESD architecture (Baño-Medina et al., 2022) as the basis for the standard DL model. This choice is motivated by several factors. First, this model is the most widely used in the downscaling literature, having been applied to various regions, including continental Europe (Baño-Medina et al., 2020, 2021), southern South America (Balmaceda-Huarte et al., 2024), Egypt (Kheir et al., 2023), New Zealand (Rampal et al., 2022), Germany (Quesada-Chacón et al., 2022), and, more recently, Iberia (Soares et al., 2023; González-Abad and Gutiérrez, 2025). Second, this model has been explicitly assessed for





its plausibility in future climate scenarios (Baño-Medina et al., 2021; González-Abad and Gutiérrez, 2025), an aspect that is often overlooked in downscaling studies (Rampal et al., 2024). Third, its promising results are achieved without having to rely on a complex or highly specialized architecture.

The DeepESD architecture employed in this work consists of three successive convolutional layers with 50, 25, and 1 kernels, respectively, each followed by a Rectified Linear Unit (ReLU) activation function (Glorot and Bengio, 2010) (see Figure 2 for a schematic overview). In the original design (Baño-Medina et al., 2020), the last convolutional layer for temperature was formed by 10 kernels instead of one. However, using 10 kernels substantially increases the network's complexity, raising the parameter count, for instance in the case of ROCIO-IBEB, from approximately 28 million (with a single final kernel) to 284

million. This increase arises from the final dense layer, which fully connects the output of the last convolutional kernel to every grid point to be downscaled. Such an overparameterized DL model may be prone to overfitting (Bishop and Nasrabadi, 2006; Hastie et al., 2001), potentially learning spurious relationships that fail to extrapolate in future scenarios, as discussed in González-Abad et al. (2023). Consequently, we choose to employ the DeepESD architecture with a single kernel in the final convolutional layer for all three variables. The output of this layer is then flattened and passed to a final dense layer, which

has as many neurons as the number of points to be downscaled, 21885 for ROCIO-IBEB and 3460 (5803) for temperature (precipitation) for STATIONS-IBEB. The differences in the latter case originate from the greater number of stations available for precipitation (see Section 2.1).

All DL models for the three variables are trained using the same procedure. We employ the Adam optimizer (Kingma, 2014) with a learning rate of $10^{-4}$ and a batch size of 64 to minimize the loss function. For temperature, we use the Mean Squared

Error (MSE), a widely adopted loss function in temperature downscaling. In contrast, selecting an appropriate loss function for precipitation is less straightforward due to its non-continuous and exponentially distributed nature. A recent study (González-Abad and Gutiérrez, 2025) found that the asymmetric (ASYM) loss function proposed in Doury et al. (2024) is well-suited for precipitation downscaling. Consequently, we use the ASYM loss function for precipitation.

To account for variability in training performance, each model is trained ten times per variable using different random

initializations of its weights. To properly evaluate the DL models, we split the dataset into training (1980–2010) and test sets (2011–2020). Additionally, during training, we set aside $10\%$ of the training data as a validation set. This validation set is used to implement an early stopping strategy with a patience of 60 epochs, ultimately selecting the model that performs best on the validation set during this period.

### 2.3 Explainable Artificial Intelligence

A key factor underlying the success of DL models is their internal structure, which involves the composition of multiple piecewise/non-linear functions and a large number of parameters. However, this complexity also makes these models difficult to interpret, often branding them as *black-boxes*, as the relationships they learn are not readily apparent. This issue is particularly relevant in statistical downscaling, where the field has transitioned from simple, interpretable models to complex, deep neural networks. To address this challenge, eXplainable Artificial Intelligence (XAI) techniques have emerged, offering insights into

the inner workings of DL models and shedding light on the relationships they capture (Adadi and Berrada, 2018; Arrieta





et al., 2020; Minh et al., 2022). Within the statistical downscaling domain, the application and benefits of XAI have only recently begun to be explored (Baño-Medina, 2021; Rampal et al., 2022; González-Abad et al., 2023; Baño-Medina et al., 2023; Balmaceda-Huarte et al., 2024).

In this study, we explore the relationships learned by the different DL models trained in the different regimes by applying the XAI-based diagnostics introduced in González-Abad et al. (2023). These diagnostics are specifically designed for the context of PP downscaling. In particular, we use the Aggregated Saliency Map (ASM), a diagnostic that quantifies the overall spatial influence of each predictor variable's grid points, thus indicating which points or variables the model focuses on most. In this work, we compute the required saliency maps by directly calculating the gradients of the predictand space with respect to the predictor space. We then apply the same preprocessing steps to these saliency maps as in González-Abad et al. (2023) before computing the ASM diagnostic.

## 3 Pre-training and Fine-tuning

Figure 2 presents a schematic view of the pre-training (left), partial fine-tuning (center), and full fine-tuning (right) regimes. For each regime, the corresponding DeepESD architecture is shown, with layers represented by small boxes indicating the number of neurons and kernel dimensions for the convolutional layers. The DeepESD architecture, across all training regimes, is divided into two distinct components: the feature extractor (depicted by the gray box in Figure 2) and the calibrator. The feature extractor, composed of convolutional layers, is responsible for learning high-level data representations. The calibrator, consisting of a dense layer, transforms these high-level features into localized predictions at each grid point of the predictand, enabling the model to perform point-by-point downscaling. Importantly, the architecture remains consistent across all three downscaled fields due to the design choice regarding the number of kernels in the final convolutional layer, as discussed in Section 2.2.

First, the DeepESD model is trained using ERA5 as the predictor and ROCIO-IBEB as the predictand, resulting in the *pre-trained* model shown in Figure 2. In this model, weights are randomly initialized, as indicated by the purple dots, and trained, as indicated by the blue arrows looping back to each layer. This corresponds to the standard training procedure for DL models. The pre-trained model then serves as the foundation for two additional variants: the *partial fine-tuned* and the *full fine-tuned* models. Both of these variants are trained using STATIONS-IBEB as the predictand, but the weights of their feature extractors are initialized using the weights learned by the pre-trained model, as shown by the purple arrows connecting the feature extractor layers across models. In the partially fine-tuned model, the feature extractor layers are frozen (i.e., not updated during training), and only the calibrator is trained, which is reflected by the absence of blue arrows for the feature extractor and its presence for the calibrator. In contrast, the fully fine-tuned model also allows the feature extractor layers to fine-tune the weights inherited from the original trained model during training.

The decision to transfer only the feature extractor originates from the role that convolutional layers play in learning high-level data representations that are often transferable across related tasks (LeCun et al., 1995; Krizhevsky et al., 2012; Agrawal et al., 2014). This is particularly relevant for the DeepESD model, where the convolutional layers capture high-level synoptic







**Figure 2.** Schematic representation of the three training regimes: pre-training (left), partial fine-tuning (center), and full-tuning (right). Each model is composed of convolutional and dense layers, grouped into two main blocks: the feature extractor (gray box) and the calibrator. Initialization is represented in purple, with points indicating random weights and arrows indicating initialization from pre-trained models. Blue looping arrows represent training. Layers with learnable parameters are shown with solid borders, while non-trainable layers are indicated with dashed borders.

patterns, while the final dense layer provides spatial specialization by fitting a linear regression over these representations for each grid point forming the predictand (Baño-Medina, 2021; González-Abad et al., 2023). Transferring only the weights of the convolutional layers (i.e., the feature extractor) aligns with standard practices in other domains, such as computer vision and natural language processing, where fine-tuning typically involves appending a task-specific layer to a trained backbone (or to the encoder in encoder-decoder architectures) (Devlin, 2018; Chen et al., 2020). The comparison between freezing these





transferred weights and fine-tuning them reflects two prevalent strategies in the pre-training literature, where some studies keep

the transferred layers fixed, while others allow them to adapt during training.

## 4    Results

In this section, we examine the feasibility of a pre-training strategy by comparing the performance and key characteristics of models originally trained on ROCIO-IBEB and fine-tuned on STATIONS-IBEB with those of a model fully trained from scratch on STATIONS-IBEB using randomly initialized weights. Throughout the manuscript, we refer to this latter approach

as *full-training*.

### 4.1    Performance of Deep Downscaling Models

Figure 3 shows the evolution of the loss function during training for the DeepESD model for each downscaled variable under the three training regimes for the STATIONS-IBEB dataset. As mentioned in Section 2.2, we use MSE as the loss for minimum and maximum temperatures, and ASYM for precipitation. For each regime, we show ten different training curves per combination,

corresponding to the independent runs with different random initializations.

Overall, the fully-trained model converges more slowly, whereas the two fine-tuned models converge significantly faster. Notably, for maximum temperature downscaling, the fine-tuned models take about half the number of epochs compared to training from scratch. For all three variables, and especially for minimum temperature and precipitation, the fully fine-tuned model achieves slightly lower training errors than the partially fine-tuned model. Regarding variability across the ten training

runs, the training dynamics for minimum and maximum temperature are highly consistent across all realizations. In contrast, precipitation shows greater variability.

In Figure 4, we present the evaluation results on the test set for the four DeepESD models: the model pre-trained on ROCIO-IBEB (pre-training), the model exclusively trained on STATIONS-IBEB (full-training), and the two models fine-tuned on STATIONS-IBEB using as foundation the pre-trained model (partial and full fine-tuning). Results are shown for minimum

(top) and maximum (bottom) temperature. For both variables, we report the Root Mean Square Error (RMSE) and the bias of the mean. Additionally, we include the bias in the annual minimum of daily minima (TNn) for minimum temperature, and the bias in the annual maximum of daily maxima (TXx) for maximum temperature. Note that the violin plots correspond to a randomly selected training run. To illustrate variability across model initializations, we include black dashed lines showing the minimum and maximum values of the spatial medians across the ten training replicas. The pre-trained model shows perfor-

mance consistent with previous studies (Soares et al., 2023; González-Abad and Gutiérrez, 2025). The three models trained on STATIONS-IBEB (fully-training, partial fine-tuning and full fine-tuning) exhibit similar performance across metrics, though with overall lower accuracy than the ROCIO-IBEB model. Notably, the partially fine-tuned model displays slightly lower variability in the spatial median, particularly for the bias in extremes (TNn and TXx), suggesting improved robustness across training runs.





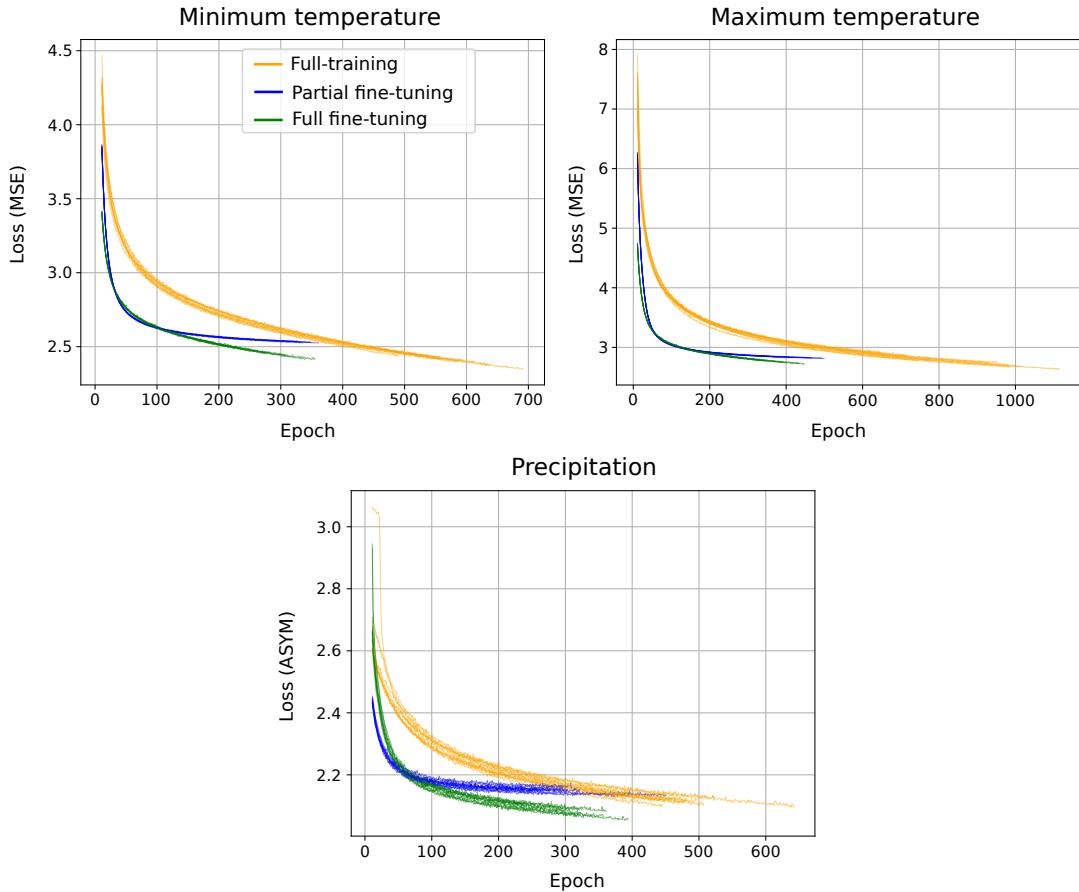

**Figure 3.** Evolution of the loss function during training for the DeepESD models across the three predictand variables (minimum temperature, maximum temperature, and precipitation) and the three training regimes (full-training, partial fine-tuning, and full fine-tuning) on the STATIONS-IBEB dataset. Each line represents one of ten training runs performed per model, using different random initializations to illustrate variability in training performance.

Figure 5 presents the equivalent analysis for precipitation. We show the Root Mean Square Error (RMSE) computed over wet days ($> 1$ mm/day), the bias in the frequency of dry days, the Simple Daily Intensity Index (SDII), and the bias in the annual maximum daily precipitation (Rx1day). As with temperature, the pre-trained model performance aligns with previous studies. The three STATIONS-IBEB models show comparable results overall, though for some metrics, the partially fine-tuned model performs slightly worse than the fully-trained and fully fine-tuned versions.

## 4.2 Explainability: Saliency of the Different Predictors

To analyze the relationships learned by the deep downscaling models, we use the XAI-based ASM diagnostic based on aggregated saliency (see Section 2.3 for details). Figure 6 displays this diagnostic computed over the test set for all predictor variables



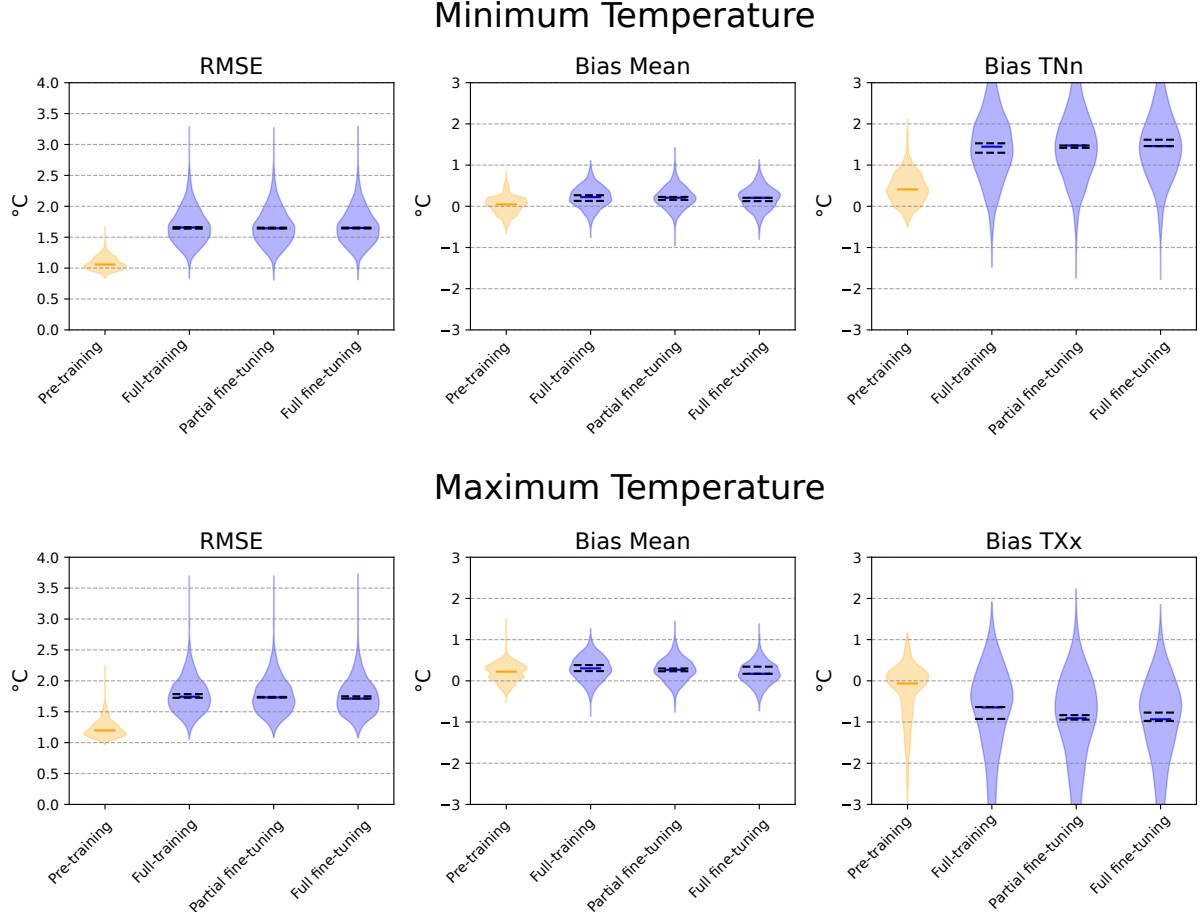

**Figure 4.** Evaluation results on the test set (2011–2020) for the DeepESD model trained under the different regimes (pre-training, full-training, partial fine-tuning and full fine-tuning). Results are shown for minimum temperature (top row) and maximum temperature (bottom row). For both variables, we report the Root Mean Square Error (RMSE) and the bias of the mean (Bias Mean). Additionally, for minimum temperature, we include the bias in the annual minimum of daily minima (TNn), and for maximum temperature, the bias in the annual maximum of daily maxima (TXx). Violin plots show the distribution across grid points for a given training run (randomly chosen), with the spatial median marked in blue. Black dashed lines indicate the range of spatial medians across the ten independent training runs.





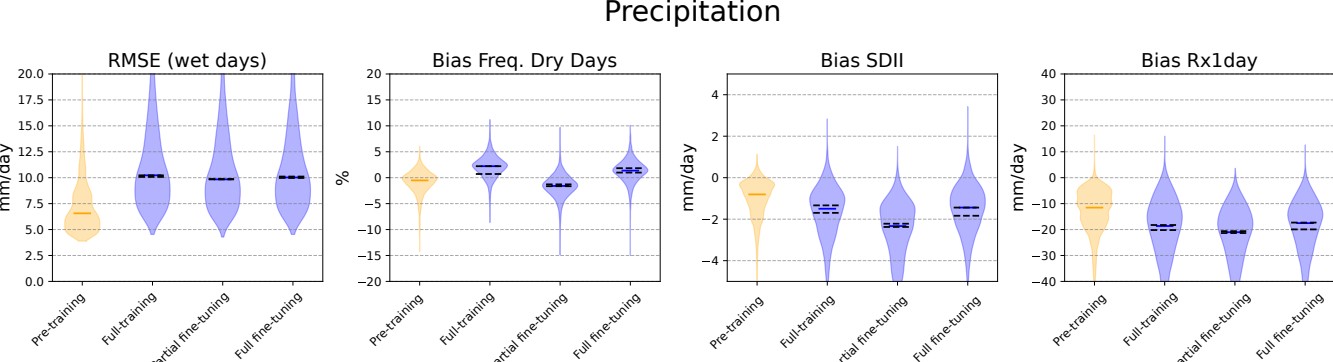

**Figure 5.** Same evaluation as in Figure 4 but for precipitation. Metrics include the Root Mean Square Error (RMSE) computed over wet days ($> 1$ mm/day), the bias in the frequency of dry days (Bias Freq. Dry Days), the bias in the Simple Daily Intensity Index (Bias SDII), and the bias in the annual maximum daily precipitation (Bias Rx1day).

for the three downscaled variables (shown in separate subplots) and for the pre-trained, fully-trained, partially fine-tuned and fully fine-tuned models. In previous work (González-Abad et al., 2023), ASM values are depicted for every grid point in each

predictor variable, thereby illustrating spatial patterns of relevance. However, to simplify our analysis, we aggregate the ASM spatially, showing a single value per predictor variable that represents its overall importance for the deep downscaling model. In Figure 6, these aggregated saliency values are displayed as histograms, with the color of each bar (gray, red, green, or blue) corresponding to the different models.

For minimum temperature, the ASM for both the pre-trained and the three STATIONS-IBEB models is distributed across

multiple variables, with air temperature and specific humidity at 850 hPa, along with mean sea level pressure, emerging as the most relevant. However, the model trained on the ROCIO-IBEB dataset (pre-training) and the STATIONS-IBEB trained model (full-training) differ in how they attribute importance to certain variables (for example, the zonal wind component at 850 hPa). This discrepancy diminishes when the weights of the feature extractor are transferred (fine-tuning), resulting in a closer alignment with the pre-trained model. Interestingly, the full fine-tuning diverges more noticeably in its ASM, showing greater

differences in variable relevance than even the fully-trained model for some predictors (e.g., mean sea level pressure). For maximum temperature, the pattern is notably different. In both the pre-trained model and the three STATIONS-IBEB models, nearly all relevance is assigned to mean sea level pressure, with only a minor contribution from air temperature at 850 hPa. In contrast, precipitation shows distributed relevance across all predictor variables, reflecting the complexity of its underlying processes. Similar to minimum temperature, certain discrepancies arise between the pre-trained and the fully-trained model

for variables such as the zonal wind component and the mean sea level pressure. These discrepancies are again resolved under either one of the two fine-tuned models, where the final ASM values converge to those of the pre-trained model.




**Figure 6.** Aggregated Saliency Map (ASM) computed over the test set for the three downscaled variables (shown in separate subplots). The model pre-trained on the ROCIO-IBEB dataset is depicted in gray, while the three models for STATIONS-IBEB (full-training, partial fine-tuning and full fine-tuning) are shown in red, green, and blue, respectively. The ASM is spatially aggregated for each predictor variable, resulting in a single importance value per variable, as represented by the bars.





## 4.3 Downscaled Climate Projections

To assess the extrapolation capabilities of the models trained under the different regimes, we compute the climate change signal for the three downscaled variables from the EC-Earth3-Veg model under the SSP3-7.0 scenario (see Section 2.1.2 for details). For temperature, this signal is obtained by subtracting the downscaled future projections (2071–2100) from those of the historical period (1980–2014), while for precipitation, it is calculated by dividing the future projections by the historical ones. Figure 7 shows the resulting climate change signals for the TNn, TXx, and Rx1Day indices, which correspond to minimum temperature, maximum temperature, and precipitation, respectively. The rows represent each index, while the columns display the signals from the original GCM as well as from the pre-trained model and the three training regimes for the STATIONS-IBEB dataset (full-training, partial fine-tuning and full fine-tuning). For the temperature-related indices (TNn and TXx), all three training regimes produce climate change signals that are broadly consistent with those of the pre-trained model. However, for TNn, these signals diverge from the climate model's output, despite showing comparable magnitudes of change. In contrast, for Rx1Day, both the pre-trained model and the three training regimes capture the climate model's signal while adding regional detail, such as reduced extreme precipitation in northern Spain. Despite this overall agreement, the fully-trained model slightly underestimates the change in the Duero River basin relative to both the pre-trained and the GCM, while the fully fine-tuned model tends to overestimate it. Only the partially fine-tuned model closely reproduces the magnitude and spatial pattern of change seen in both the GCM and the pre-trained model.

To assess the variability of the resulting climate change projections, we train the DeepESD model ten times under each of the three different regimes, changing only the random initialization of the parameters. In the fully-trained model, all parameters are randomly reinitialized for each replica, whereas in the two fine-tuning regimes, only the parameters of the final dense layer are reinitialized. Figure 8 shows the standard deviation of the climate change signals for the TNn, TXx, and Rx1Day indices (rows) across model replicas for each training regime (columns). As expected, the fully-trained model exhibits the highest variability as no parameters are fixed, increasing the sources of variation across replicas. In contrast, although the fully fine-tuned model updates all parameters, it begins each training with the same initialized feature extractor, reducing variability across replicas. This effect is even more pronounced in the partially fine-tuned model, where the feature extractor is kept fixed and only the dense layer is trained. For temperature indices, both fine-tuning regimes show similar low variability. However, for precipitation, the fully fine-tuned model shows greater variability than the partially pre-trained one, particularly in southeastern Spain.

## 4.4 Sensitivity to data record length

Finally, to evaluate the benefits of relying on pre-trained models in low-data regimes, we train the aforementioned DL models on different versions of the STATIONS-IBEB dataset, each featuring a distinct proportion of missing data points. These versions are created by randomly removing observations across both space and time. During training, the DL models update their parameters only taking into account the non-missing data points of the predictand, which enables them to learn even when





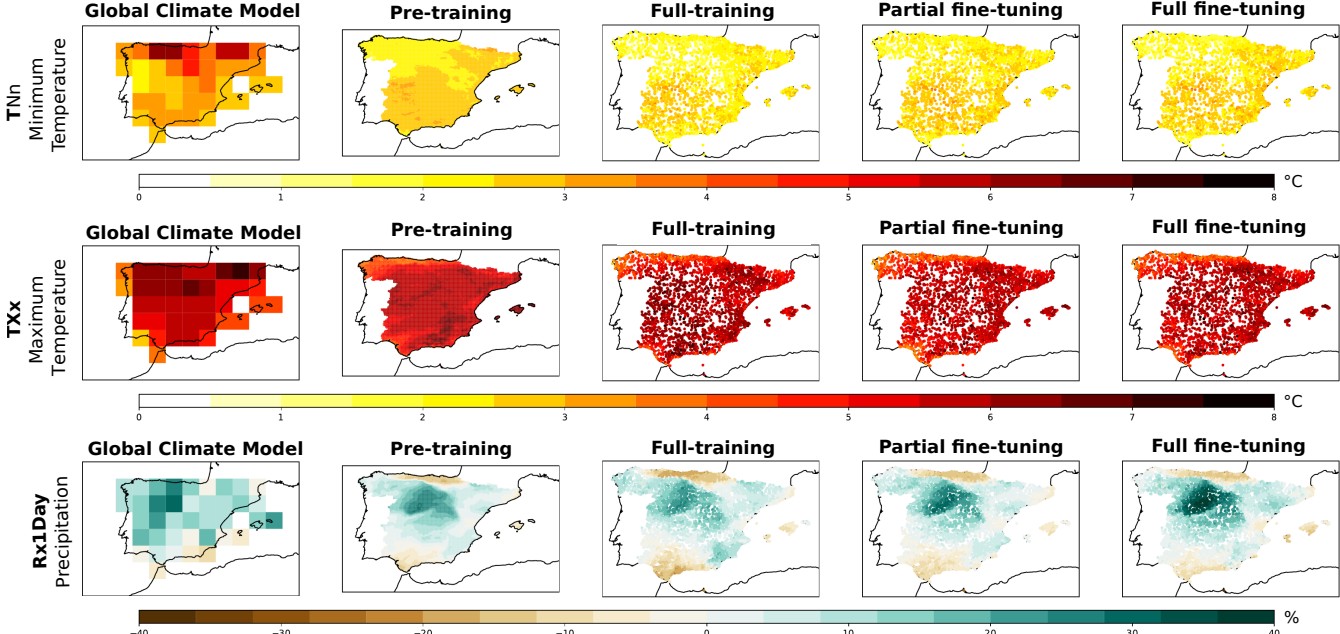

**Figure 7.** Climate change signals corresponding to the EC-Earth3-Veg model for the annual minimum of daily minimum temperatures (TNn), the annual maximum of daily maximum temperatures (TXx), and the annual maximum daily precipitation (RX1day). Each index is shown in a separate row, displaying the climate model's signal as well as the downscaled signals from the pre-trained model and the three different training regimes for the STATIONS-IBEB dataset (fully-training, partial fine-tuning and full fine-tuning), in columns. Note that the temperature signals are expressed in $^{\circ}C$, whereas precipitation signals are expressed as percentages.

portions of the dataset are missing. Despite this partial data availability, the model retains the same architecture as the one trained without missing points, and thus can still generate predictions across the entire predictand domain.

In Figure 9, we present the Root Mean Squared Error (RMSE) on the test set for the three downscaled variables under the different training regimes. Each regime is trained on versions of the ROCIO-IBEB dataset with varying percentages of artificially introduced missing values, shown on the x-axis of each subplot. Note that the RMSE is computed on the test partition without missing data, enabling us to evaluate the model's ability to extrapolate to data points that were partially missing during training. Overall, fine-tuning leads to better performance in the presence of missing data in the STATIONS-IBEB dataset. For both minimum and maximum temperatures, the fine-tuned regimes achieve lower RMSE values than the fully-trained regime, converging to the same value when the training is carried out on the STATIONS-IBEB dataset without missing data. The advantage of fine-tuning is especially pronounced under high missing-data regimes (e.g., $90\%$), where having pre-learned representations aids in generalizing to unseen data points. For precipitation, a similar trend is observed between the fully-training and both fine-tuning regimes; however, the fully fine-tuned model performs worse than the others when more than $60\%$ of the training data is missing, though it still outperforms the fully-trained model beyond this threshold.





**Figure 8.** Standard deviation of the climate change signals for the TNn, TXx, and RX1Day indices (shown in rows), computed across ten training replicas under the fully-training, partial fine-tuning and full fine-tuning regimes. The standard deviation for temperature signals is expressed in $^{\circ}C$, while that for precipitation signals is given as a percentage.

## 5 Discussion

The results presented in this work demonstrate the potential of pre-training for PP downscaling, particularly for developing a standard model in the region of Spain. Specifically, as shown in Section 4.1, fine-tuned DeepESD models show accelerated
convergence, indicating that representations learned on the ROCIO-IBEB dataset contain valuable information for modeling the point-based dataset. However, this benefit does not necessarily translate into improved accuracy on STATIONS-IBEB, likely due to the presence of higher and more localized extreme values, which are more challenging to model than their smoothed counterparts in the interpolated ROCIO-IBEB gridded dataset. In the case of precipitation, fine-tuning the feature





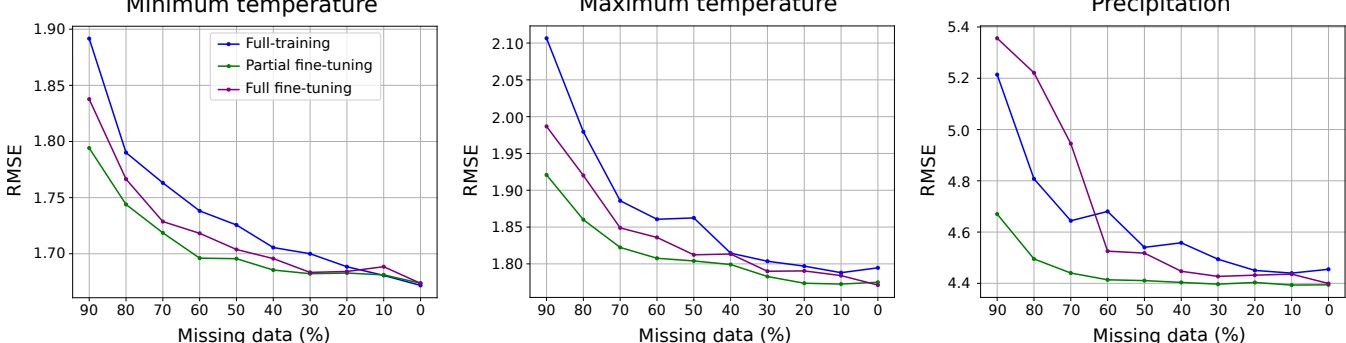

**Figure 9.** Root Mean Squared Error (RMSE) computed over the test set for three different training regimes (fully-training, partial fine-tuning and full fine-tuning), shown for the three downscaled variables (minimum temperature, maximum temperature and precipitation), in columns. Each model is trained on a different version of the ROCIO-IBEB dataset, varying in the proportion of artificially introduced missing data (indicated on the x-axis of each subplot). Note that the RMSE shown here is computed using the ROCIO-IBEB test partition without missing data.

extractor appears to be beneficial, indicating that for this variable, slight adjustments to the learned large-scale patterns may be
necessary to better capture local characteristics.

    When analyzing the patterns learned by the various models, by focusing on the ASM diagnostic, we find differences in the distribution of relevance across predictors for the three downscaled variables. Focusing on minimum temperature, all models exhibit their highest relevance in air temperature and specific humidity at 850 hPa (the lowest level) and mean sea level pressure, although the fully-trained model also shows a relatively high relevance for the zonal wind component at 850 hPa. This finding
aligns with previous studies (Baño-Medina, 2021; González-Abad et al., 2023), which indicate that, for temperature, the most critical variables are air temperature and specific humidity at 1000 hPa, alongside geopotential height at 1000 hPa—closely related to mean sea-level pressure in our setup. The fully-trained and fully fine-tuned models show the greatest deviation from the expected relevance distribution, particularly in the increased ASM assigned to mean sea level pressure by the fully fine-tuned model. Nonetheless, the overall relevance patterns remain broadly similar. In contrast, the partial fine-tuned model
exhibits relevance distributions more closely aligned with those of the pre-trained model, suggesting that, for this variable, fine-tuning the feature extractor may introduce a divergence from the relationships learned by the pre-trained model.

    The ASM diagnostic indicates a stronger relationship between mean sea level pressure and maximum temperature than with minimum temperature. This is consistent with our understanding of temperature variability. Maximum temperature is largely controlled by atmospheric circulation patterns and adiabatic warming, both of which are effectively captured by this predictor.
In contrast, minimum temperature typically requires a more complex set of predictors due to the influence of local surface conditions, moisture availability and radiative cooling processes. This distinction aligns with the fundamental meteorological principles governing daytime heating and nighttime cooling, as well as with previous studies (Favà et al., 2016, 2018; Merino et al., 2018; Pérez and García, 2023) that have demonstrated a robust relationship between pressure-related variables and tem-





perature variability over the Iberian Peninsula. Interestingly, recent work (Cariou et al., 2025) has shown that DL models can

effectively reproduce daily temperature variations over Europe using only mean sea level pressure as input, thereby underscoring the strong influence of atmospheric circulation. This aligns with our XAI-based findings, which highlight the particular relevance of mean sea level pressure, especially for maximum temperature.

In the case of precipitation, relevance is distributed across the full set of predictor variables, which aligns with the fact that precipitation dynamics encompass a broader range of phenomena. As shown in Figure 6, wind components play a significant

role, particularly in northwestern regions, where westerly winds transport moisture from the Atlantic Ocean. The influence of both temperature and humidity also makes sense in this context; for example, higher temperatures require greater amounts of humidity to reach saturation, a key factor in precipitation events. Additionally, mean sea-level pressure is relevant because pressure differentials can trigger the lifting of moist air. Overall, while all training regimes agree on the distribution of relevance, the partially fine-tuned model produces values that more closely align with those of the pre-trained model.

Regarding the climate change signals produced by these models for minimum and maximum temperature, the results closely align with those reported in González-Abad and Gutiérrez (2025), indicating plausible projections. The lack of relevant differences among the signals from the different training regimes may originate from the use of 30-year climatologies, suggesting that the choice between the ROCIO-IBEB and STATIONS-IBEB datasets does not fundamentally impact statistical trends at these temporal scales. As expected, the partially fine-tuned model exhibit lower variability than the fully fine-tuned one since

fixing a subset of parameters constrains the optimization process, limiting it to a smaller region of the parameter space and leading to convergence within similar local minima (Erhan et al., 2010). Overall, if the pre-trained model is deemed reliable, fine-tuning strategies can help reduce epistemic uncertainty, reinforcing confidence in the projections.

As shown in Section 4.4, pre-training is beneficial in low-data regimes, as it provides new DL models with useful representations. This advantage has been previously observed in the SR field (Zhu and Zhou, 2024), where pre-trained models have

successfully transferred relationships across different regions. However, in SR, regional transferability is more straightforward since the model learns an interpolation function by relying on a coarser version of the predictand. In contrast, in the PP approach, the empirical relationships being learned may depend on dynamic regional phenomena, making them potentially less transferable across different regions.

## 6 Conclusions

Current statistical downscaling methodologies often involve different research groups employing various schemes and drawing on datasets of varying spatial resolution and temporal coverage; this can lead to inconsistencies in the resulting downscaled projections. Such discrepancies pose challenges when nationwide products exist—for example, Escenarios-PNACC 2024 in Spain—as discrepancies may arise. To address this issue, we introduce pre-training as a strategy to develop a standard DL model for PP downscaling in Spain, ensuring consistency when adapting it to different regional datasets. Specifically, we

propose a standard model by using as the basis the DeepESD architecture trained on the ROCIO-IBEB gridded dataset. We



then demonstrate how this model can serve as a foundation for training on a point-based dataset (STATIONS-IBEB) using two approaches: partial and full fine-tuning.

As demonstrated by our XAI-based analysis, the pre-training paradigm enables the resulting model to inherit the relationships learned by the pre-trained model, grounding its climate projections in the same physical basis. In the case of Spain,

these inherited relationships align with previous findings in the literature (Baño-Medina, 2021; González-Abad et al., 2023), and the fine-tuned versions follow suit. Such inheritance is particularly promising when transferring to small regional datasets, where DL models are prone to overfitting and learning spurious patterns (González-Abad et al., 2023). Moreover, leveraging an established downscaling model reduces the epistemic uncertainty of the fine-tuned model, as the parameter search space is constrained to that of the model taken as the basis. When this model is proven to be reliable, this reduction in uncertainty

further enhances trust in the final projections. Beyond ensuring greater consistency with this model, our work suggests that pre-training offers additional benefits in the context of PP downscaling. For instance, if the datasets used for the pre-trained and the fine-tuned models share certain characteristics (e.g., a similar geographic region), the fine-tuned model converges faster by effectively leveraging the representations learned by the pre-trained model. Furthermore, in low-data regimes—where datasets often have missing observations due to station errors or other issues—fine-tuning allows the DL model to rely on the already

learned relationships. This is especially valuable in regional contexts, where incomplete or inconsistent datasets are common.

The promising results of pre-training in the context of PP downscaling open several avenues for future research. In the case of Spain, one potential approach to creating a more representative basis model is to leverage the range of available observational datasets (Peral García et al., 2017; Herrera et al., 2019), which are generated using various methods (e.g., different interpolation techniques). Integrating multiple datasets could help reduce observational uncertainty tied to dataset selection.

Another promising direction is to extend this approach to larger regions, such as Europe, where multiple observational datasets are also available. In such continental regions, downscaling models are typically trained on continental or global observational datasets that may not reflect the specific properties of individual regions (see, e.g., Baño-Medina et al. (2022)). In this case, a standard model could be trained on broader datasets and then transferred with regional datasets at higher resolutions, further enhancing its accuracy and applicability in generating nationwide products. Lastly, following efforts both within (Lessig et al.,

2023; Nguyen et al., 2023; Bodnar et al., 2024) and beyond (Bommasani et al., 2021) the climate and weather domains, a self-supervised learning strategy could be employed to train a foundational model by drawing on data from GCMs and Regional Climate Models (RCMs) (Rampal et al., 2024). Such a foundational model could then serve as a basis for numerous regions, variables, and even different downscaling tasks (e.g., PP downscaling or emulation).

Overall, pre-training emerges as a promising strategy for developing a standard PP downscaling model, particularly for

Spain, the focus of this study. By fine-tuning a pre-trained model, we can retain its physical basis, ensuring consistent projections and reducing epistemic uncertainty in critical extrapolation scenarios, such as downscaling GCMs under future climate conditions. Moreover, pre-training confers additional benefits, including enhanced generalization in low-data regimes. These advantages position pre-training as a valuable avenue for future research, potentially enabling robust pre-trained models that a wider community of users and researchers can readily adopt.



*Code and data availability.* All the code and data necessary to reproduce the experiments in this study are publicly available. The pro-
cessed ERA5 dataset and the data from EC-Earth3-Veg are freely accessible via a Zenodo repository (https://zenodo.org/records/16687087,
González-Abad 2025c). The ROCIO-IBEB dataset is available on the AEMET website (https://www.aemet.es/en/serviciosclimaticos/cambio_
climat/datos_diarios?w=2), as is the STATIONS-IBEB dataset (https://archivo-proyecciones-climaticas.aemet.es/). The code to fully repro-
duce the experiments is publicly accessible (https://zenodo.org/records/16680731, González-Abad 2025a). Additionally, all trained models
used to generate the results are provided in a Zenodo repository (https://zenodo.org/records/16681330, González-Abad 2025b), ensuring the
exact reproducibility of the findings presented in this manuscript.

*Author contributions.* The conceptualization of the study was carried out by JGA and JMG. Code development was performed by JGA, while
data acquisition and processing were conducted by JGA, MI and AH. Formal analysis was undertaken by JGA and JMG, and visualization
was prepared by JGA. All authors contributed to the writing, reviewing, and editing of the manuscript.

*Competing interests.* The authors declare that they have no conflict of interest.

*Acknowledgements.* We would like to acknowledge all the teams involved in the production and maintenance of the ERA5 reanalysis dataset
and the EC-Earth3 climate model simulations. We express our special gratitude to the teams at AEMET for the development and provision
of the ROCIO-IBEB and STATIONS-IBEB datasets used in this work. González-Abad acknowledges support from grant CPP2021-008510
funded by MICIU/AEI/10.13039/501100011033 and by the "European Union" and the "European Union NextGenerationEU/PRTR".



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
