# Peer review of "Pre-training for Deep Statistical Climate Downscaling: A case study within the Spanish National Adaptation Plan (PNACC)"

_EGUsphere, 2025_

## Author Comment (AC2)

*We thank the reviewer for the thorough and insightful review. The referee correctly identified some methodological limitations, and provided concrete suggestions that have been very helpful in planning the revised paper. Below we respond to the overall assessment and all major comments regarding the future revision of the manuscript.*

**In this study, the authors aim to explore the benefits of transfer learning in the context of ML-based statistical downscaling for climate applications. Specifically, the goal is to understand the benefits of pre-training the latent representations of ML-based downscaling models on a core dataset, to achieve improved skill or higher consistency when the downscaling model is fine-tuned for a different downscaling task.**

*Thank you for your general comment. In the revision we will better clarify and motivate the main objective of the paper, to enhance consistency across downscaling models trained on different observational references (e.g. national and regional wide). This is particularly relevant for the provision of national and regional climate change scenarios with multiple (national and regional) datasets, illustrated in this paper with the example of the regional climate change scenarios produced for the Spanish Adaptation Plan (PNACC). In these cases, coherent climate-change signals across products are essential for downstream use. We will revise the abstract and introduction accordingly, and we will add experiments on an independent station-based target dataset (unrelated to ROCIO-IBEB) to test whether these consistency/alignment benefits persist when the fine-tuning data are not closely related to the pre-training dataset (see responses to the related major comments below).*

**This topic is certainly of practical and scientific interest for climate modelers, and a good fit for the journal. However, I find the implementation of this study to be largely uninformative regarding the questions posed by the authors in the abstract and introduction. In my view, this is due to the choices made by the authors for the pre-training and fine-tuning datasets, as well as the forms of fine-tuning explored for the DeepESD model. For these reasons, which I detail below, I find this manuscript in its current form unsuitable for publication. I encourage the authors to find a more effective lens through which the questions that they set out to study can be answered.**

*In order to address the concerns of the reviewers, we have used an additional (regional) fine-tuning dataset allowing us to better illustrate the main results of the paper. We will include the results with this new example in the revised version to better support the questions raised in the abstract and introduction.Detailed point-by-point responses are provided below under each major comment.*

**Major comments:**

**- Transfer learning is a well-established way to fine-tune large ML models, pre-trained on an extensive pre-existing dataset, on a smaller dataset that is more representative of some final task. The manuscript instead explores pre-training roughly 17,000 parameters of the final DeepESD models, out of a total of ~4.4M parameters for temperature and ~7.5M parameters for precipitation, respectively. This can hardly be called fine-tuning, when the pre-trained parameters represent less than 1% of the total**

**number of parameters in both cases. This partly explains why all the variants yield statistically equivalent results (Figures 4-7), and why no conclusions can be drawn from this experimental setup.**

*We agree that, in our current DeepESD configuration, the number of parameters transferred from ROCIO-IBEB is small compared to the full model. At the same time, we would like to emphasize that parameter count alone is not a reliable proxy for "functional" importance. In the DeepESD model, the feature extractor may contain relatively few parameters due to parameter sharing, yet it is the component that learns the reusable, nonlinear representations that determine what information is available to the final fine-tuned layer [1].*

*In DeepESD, the calibrator (final dense layer) contains many parameters primarily because it scales with the number of output locations, but, given the high-level representations, it acts as a location-dependent linear mapping from those learned representations to the final prediction. Transfer via a fixed feature extractor plus a liner mapping is also a common paradigm in the transfer-learning literature. In addition, prior work shows that transferability depends strongly on layer role/depth, with earlier representation layers often being more transferable than later task-specific layers [2].*

*Consistent with this interpretation, our results (Figure 8) show that when the feature extractor is frozen and only the calibrator is fine-tuned, the spread across replicas becomes nearly zero, suggesting that, once representations are fixed, tuning the calibrator has limited impact. We will expand the discussion of this point in the revised manuscript and include the above references (among others).*

*[1] Goodfellow, I., Bengio, Y., Courville, A., & Bengio, Y. (2016). Deep learning (Vol. 1, No. 2). Cambridge: MIT press.*

*[2] Yosinski, J., Clune, J., Bengio, Y., & Lipson, H. (2014). How transferable are features in deep neural networks?. Advances in neural information processing systems, 27.*

**- The final target dataset of interest, STATIONS-IBEB, is used to construct the pre-training dataset, ROCIO-IBEB. This setup omits the most important practical aspect of transfer learning: can we train on one dataset to improve predictive skill on a different dataset? This is an important question for some of the applications cited by the authors: pre-train on a national-level dataset, and fine-tune on a local dataset with different statistics (Taboada et al, 2024). The current setup is too idealized and not representative of the situations where transfer learning may actually be useful, in my opinion.**

*We agree with the reviewer that our original ROCIO-IBEB/STATIONS-IBEB setup is not an ideal test of the practical transfer-learning question, because the gridded pre-training target is derived (at least in part) from the same underlying observational network. To address this directly, in the revised manuscript we add additional experiments based on an independent station dataset over Cataluña region (not overlapping with STATIONS-IBEB and not involved in the construction of ROCIO-IBEB). This setting more closely matches the intended use case highlighted by the reviewer: pre-train on a broader "core" dataset and fine-tune on a local dataset with different statistics and limited coverage.*

*Concretely, we pre-train the model on ROCIO-IBEB and fine-tune it on the Cataluña station dataset, and we compare against training-from-scratch baselines under identical predictors, splits, and evaluation metrics. The full analysis and discussion will be included in the revised manuscript. Preliminary results indicate that the main qualitative findings of the study (in particular, the robustness/consistency benefits of leveraging a common pre-trained representation in data-sparse settings) persist in this fully independent transfer scenario, and we will revise the conclusions accordingly.*

**- Figure 3 shows that pre-training does not yield improved results, only faster training for a relatively inexpensive model where training cost is not really an issue. The rest of the results also fail to show any positive effects of pre-training. I think this is because at the level of the high-level representations of the data learned by the convolutional layers, the datasets ROCIO-IBEB and STATIONS-IBEB are largely indistinguishable (since they share the same data sources). This leads me to believe that the improved skill of fine-tuned models when the STATIONS-IBEB dataset is artificially shrunk (Fig 9) is due to the fact that you are actually showing a very similar version of the omitted samples to the model through ROCIO-IBEB. This is another example of why useful conclusions cannot be drawn given the similarity of the two datasets considered.**

*We agree that Figure 3, as currently discussed, should not be interpreted as evidence of improved final skill. In the revision we will reposition it as evidence that the pre-trained high-level representations are useful for the fine-tuned task, as they allow the model to converge faster than training from scratch while keeping the same interpretability (importance of predictors) of the pre-trained model, thus producing consistent models for national and regional climate change projections*

*Regarding the concern about ROCIO-IBEB and STATIONS-IBEB being too similar, we agree this limits the conclusions that can be drawn from that pairing. This is precisely why, in the revised manuscript, we add the independent Cataluña station dataset experiment and refine the framing/objectives (including the PNACC motivation), to demonstrate the usefulness of the approach in a more realistic transfer setting with sparse data.*

**Minor comments:**

**- L38-40: "Diverging outcomes, which may confuse users". Improving consistency at the expense of capturing the true uncertainty of regional climate projections is actually a disservice to the downstream users, because it leads to biased estimates of risk.**

**- L82: Calling a 1km resolution downscaled dataset spanning thousands of years and supported by extremely sparse observations is a stretch (Karger et al, 2023).**

**- L123: "the most widely used in the downscaling literature". I don't think this method is that well established (60 studies reference it), so this needs to be toned down. There are studies from 2024 on downscaling with more references (e.g., CorrDiff), and**

deterministic ML-based downscaling models are not representative of the state of the art anymore.

- The version of DeepESD used in this paper is different than the one introduced by Bano-Medina et al (2022) for temperature. The MSE loss assumes a homogeneous uncertainty estimate, unlike the original Gaussian log-likelihood where the variance is explicitly modeled. I would also say that the deterministic MSE is no longer a "widely adopted" loss in downscaling due to its tendency to smooth fields in space and underestimate extremes.

- Fig 2: The legend reads "full-tuning" for the right column It should be full fine-tuning.

- Fig 3: Is this the training loss or the validation loss? If the former, please change to the validation loss, which is more representative of operational skill. Otherwise, please show both.

- L212: "take about half the number of epochs": Certainly not to reach the best final skill, since the fully trained model is better. How are you defining a common final time for all models to assert this?

- The results shown in Figures 4 and 5 for the ROCIO-IBEB dataset are not comparable to those in the STATIONS-IBEB dataset, since the former is a smoothed interpolation of the latter. Errors on the ROCIO-IBEB dataset will always be lower.

- Figure 9: I think the legend should refer to different versions of STATIONS-IBEB, not ROCIO-IBEB.

- Discussion: I do not know where the study demonstrates "the potential of pre-training" (L303), or the affirmation that "fine-tuning the extractor appears to be beneficial". Beyond Figure 9, which has some issues I raised before, the other results are largely equivalent for all variants.

---

## Author Comment (AC3)

*We thank the reviewer for the careful and constructive review. Their comments clearly identified key limitations and provided guidance that has helped us strengthen the revised manuscript. Below we address the overall assessment and all major points raised.*

**This study presents an important exploration of using pre-trained deep learning (DL) models for climate downscaling, aiming to maintain physical consistency between large-scale predictors and localized datasets. By systematically testing multiple training strategies (pre-training, partial fine-tuning, full fine-tuning, and full training), the authors demonstrate the robustness and efficiency of applying pre-trained models on the station-based dataset. However, as the authors note in the discussion, "this benefit does not necessarily translate into improved accuracy on STATIONS-IBEB, likely due to the presence of higher and more localized extreme values, which are more challenging to model than their smoothed counterparts in the interpolated ROCIO-IBEB gridded dataset." This observation raises a critical issue: while pre-training improves efficiency and generalization, it may limit the model's ability to capture localized extremes that define station-based observations. Clarifying this trade-off would deepen the study's insight into how pre-trained DL models balance physical consistency and predictive reliability in downscaling applications. The following comments aim to clarify and deepen several aspects of this discussion.**

*We thank the reviewer for highlighting this important trade-off. We agree that pre-training may not necessarily translate into improved skill on STATIONS-IBEB, and clarifying this balance is central. In our study, the primary objective is not to maximize accuracy, but to assess whether pre-training can promote consistency across products and align the learned large-scale dependencies of models trained on different (often sparse) observational targets (we acknowledge that this objective was not stated clearly enough in the current manuscript). Importantly, our results indicate that fine-tuning delivers these benefits without degrading overall skill compared to training from scratch, while improving training efficiency and robustness. We will revise the manuscript to make these objectives explicit and to better frame the consistency-accuracy trade-off in the discussion.*

**The key distinction between ROCIO-IBEB and STATION-IBEB lies in their treatment of local extremes. Since station-based datasets inherently preserve localized weather phenomena, it would be helpful to elaborate on the rationale for using STATION-IBEB as the downscaling target and to contrast its statistical characteristics—particularly the distribution tails representing extreme events—with those of ROCIO-IBEB. This clarification would highlight the physical implications of transferring knowledge between datasets with distinct spatial and statistical properties.**

*We agree. In the revised manuscript we will strengthen the motivation for using STATIONS-IBEB as the target: unlike the gridded ROCIO-IBEB product, the station network preserves local variability and extremes, which is precisely the challenging regime we want to test when moving from a "core" product to point-scale observations. We will apply the same characterization to the additional independent station dataset we will introduce in the revision.*

**Although fine-tuned models converge faster and achieve comparable performance to fully-trained models in terms of RMSE and mean bias, Figure 4 (right column) suggests that fully-trained models perform slightly better for extreme metrics such as TXx and TNn. This raises an important question about the ability of pre-trained models to represent localized extremes, which are critical for reliable high-impact weather downscaling. A focused evaluation of model skill over the extreme subsets of both ROCIO-IBEB and STATION-IBEB would help determine whether performance limitations stem from the coarse representation of extremes in the pre-training data or from the fine-tuning process itself, which may not fully adapt to station-scale variability.**

*We thank the reviewer for highlighting this important point. We agree that, in principle, fine-tuning from a model pre-trained on a smoother gridded product could limit the ability to fully adapt to the more localized extremes present in station-based targets such as STATIONS-IBEB, and the current results may be an early indication of this trade-off. In the revised manuscript we will examine this more directly by expanding the evaluation to focus on extreme indices and/or extreme subsets and leveraging the additional independent station-based dataset to assess whether any limitation in representing extremes is systematic across station targets. If present, we will incorporate these findings into the discussion as a clearly stated limitation and as guidance for future extensions of the approach.*

**The aggregated saliency map results reveal differences between full-training and pre-trained models, yet it is unclear whether these reflect meaningful large-scale dependencies capable of inferring local extremes or potential overfitting to dominant features. Providing examples of regional or event-specific saliency maps, rather than only aggregated values, would clarify whether the learned features correspond to physically interpretable meteorological patterns or spurious correlations introduced during training.**

*We agree with the reviewer. In the current manuscript we mainly report aggregated saliency statistics, and while the resulting predictor-importance ranking is consistent with what has been reported in related downscaling/XAI studies [1], this does not by itself demonstrate that the learned dependencies are physically meaningful at the event/regional scale. In the revised manuscript we will consider adding examples of regional/event-specific saliency maps to assess whether the highlighted patterns are meteorologically interpretable rather than driven by dominant but spurious features. We will also extend the same XAI analysis to the additional independent station dataset introduced in the revision, to examine whether fine-tuning consistently promotes alignment of large-scale dependencies across different station targets (as motivated in our responses above).*

*[1] González‑Abad, J., Baño‑Medina, J., & Gutiérrez, J. M. (2023). Using explainability to inform statistical downscaling based on deep learning beyond standard validation approaches. Journal of Advances in Modeling Earth Systems, 15(11), e2023MS003641.*

**In Section 4.4, fine-tuned models trained with datasets containing varying fractions of missing data show lower RMSE values, attributed to pre-learned representations improving generalization. However, such pre-training could potentially smooth out**

**localized extremes in unseen data. Evaluating performance specifically under extreme conditions in these incomplete datasets would strengthen the interpretation of how pre-training affects robustness and physical fidelity when data coverage is limited.**

We agree. In the revised manuscript we will extend the analysis in Section 4.4 to better account for extreme behavior under missing data, in line with the additional extremes-focused evaluation described in our response to a previous comment. This will help clarify whether the robustness gains we observe with pre-training are achieved without compromising the representation of localized extremes when data coverage is limited.

**Overall, this study makes a valuable contribution to understanding how pre-trained DL models can be adapted for regional climate applications. Further analysis focusing on extreme events and saliency-based interpretability would enhance confidence in the approach and clarify the trade-offs between maintaining physical consistency and capturing localized, high-impact weather phenomena.**